# Multi-level factors influencing HIV risk behaviors and oral PrEP use among Black and Latino men with heterosexual contact in New York City

Ohshue S. Gatanaga[1]*, Dalila Victoria Lanza[2], Robert A. Pitts[3], Ronald S. Braithwaite[2,4], Sahnah Lim[2]

1 Department of Sociomedical Sciences, Columbia University Mailman School of Public Health, New York, New York, United States of America, 2 Department of Population Health, New York University Grossman School of Medicine, New York, New York, United States of America, 3 Department of Medicine, New York University Grossman School of Medicine, New York, New York, United States of America, 4 Department of Medicine, New York University Grossman School of Medicine, New York, New York, United States of America

* osg2105@columbia.edu

## Abstract

### Background

New HIV infections are increasing among heterosexual Black and Latino populations in the United States, yet little is known about the shared characteristics of HIV risk behaviors and oral PrEP use among these populations. This study sought to understand factors influencing HIV risk behaviors and oral PrEP use among low income, limited English proficient (LEP), Black and Latino men with heterosexual contact in New York City (NYC).

### Methods

Sixteen Black and Latino cisgender men with heterosexual contact and indication for PrEP were recruited from an urban safety net hospital in NYC between 2021 and 2022. In-depth interviews were conducted with participants in English and Spanish. Thematic content analysis was conducted.

### Results

Participants described multi-faceted experiences around PrEP use informed by HIV stigma, limited understanding of PrEP, and PrEP attitudes from participants and their social networks. Participants' HIV risk behaviors were informed by sexual decision making rooted in hegemonic masculinity, trust, and perceived HIV and STI susceptibility. Participants disclosed the mixed and diverse nature of their sexual networks (i.e., cisgender women, gay men, transgender women, and sex workers). Among LEP Spanish-speaking participants, HIV risk behaviors were contextually embedded in histories of sexual or physical abuse, child labor, and/or substance use problems.

**Editor:** Sebastian Suarez Fuller, University of Oxford Nuffield Department of Clinical Medicine: University of Oxford Nuffield Department of Medicine, UNITED KINGDOM OF GREAT BRITAIN AND NORTHERN IRELAND

**Data availability statement:** Data cannot be shared publicly because of these are individual in-depth qualitative interviews (N=16). Participants share highly sensitive information such as disclosure of child sex and physical abuse. Data are available from the NYU Grossman School of Medicine IRB (contact via irb-info@nyulangone.org) for researchers who meet the criteria for access to confidential data.

**Funding:** This work was supported by Gilead Sciences Inc under Grant IN-US-276-5456 and by the NIH – National Heart, Lung, and Blood Institute [grant number 1R01HL160324].

**Competing interests:** The authors have declared that no competing interests exist.

## Conclusions

Our results call for interventions that improve sexual health knowledge, reduce stigma, and foster open discussions around sexual networks. Combined, these interventions may contribute to more accurate understanding of HIV risk behaviors and reduction of HIV transmission, especially among communities impacted by sociopolitical disenfranchisement such as LEP Spanish-speakers.

## Introduction

Human immunodeficiency virus (HIV) disproportionately impacts Black and Latino/a populations in the United States (U.S.), who in 2019 respectively comprised 40.3% and 24.7% of the new HIV diagnoses despite accounting for 13.4% and 18.5% of the population [1]. While extant HIV research in the U.S. context has largely focused on men who have sex with men (MSM) and people who inject drugs (PWID), the magnitude of racial and ethnic disparities in HIV diagnosis remains more pronounced among heterosexual individuals than in MSM and PWID [2]. These disparities were particularly apparent among Black heterosexual men, who were 29 times more likely to be diagnosed with HIV than their white counterparts in 2015 [2]. In New York City (NYC), over half of diagnosed HIV cases were linked to heterosexual contact in 2021, indicating notable risk of HIV transmission among both heterosexual men and women [3].

Black and Latino heterosexual men may be at heightened risk for HIV transmission related to HIV risk behaviors, and limited understanding of PrEP. Stressors such as mass incarceration and fewer opportunities for socioeconomic advancement in historically-disenfranchised, urban neighborhoods have been associated with increased HIV transmission rates among Black heterosexual men [4]. Low levels of sexual health literacy among Black and Latino immigrants stemming from stigma around discussing sex and limited access to multilingual resources may also contribute to increases in HIV-risk behaviors [5]. Additionally, ideologies around hegemonic masculinity and gendered power relations in Black and Latino communities contribute to HIV risk behaviors such as inconsistent condom use and multiple sexual partners among heterosexual men [6]. Hegemonic masculinity posits that there exists a set of societal norms, often established by men in power to organize gendered dynamics in unequal ways, that reinforces men's domination over women as well as the power of some men over other men [7]. As such, HIV risk behaviors driven by normative expectations around the subordination of women are especially salient in relational contexts and may contribute to increased HIV transmission rates, and gender inequity at both the personal and societal level.

Low uptake of oral pre-exposure prophylaxis (PrEP), a highly effective medication taken daily to prevent HIV acquisition among HIV-negative individuals, may also contribute to higher risk of HIV transmission among Black and Latino/a populations [8]. In 2015, the World Health Organization and Centers for Disease Control and Prevention (CDC) updated their guidelines to include heterosexually active adults as

a population that could greatly benefit from PrEP usage [9,10]. In the same year, the CDC recommended the administration of PrEP to 624,000 heterosexual individuals in the U.S. with indications for PrEP [10]. However, 2019 data from the CDC National HIV Behavioral Surveillance found that 32.3% of heterosexually active adults were aware of PrEP, but less than 1% of heterosexually active adults in high-prevalence cities utilized PrEP—a finding that was particularly pronounced among Hispanic or Latino men and limited English proficient (LEP) individuals [11]. Uptake among heterosexual men remains especially low, a finding reflected in various community studies across urban settings in the U.S. [12].

Despite the documented risk factors for HIV transmission and lower PrEP uptake among Black and Latino heterosexual men, few studies have examined factors influencing HIV risk behaviors and PrEP uptake concurrently, in spite of the shared characteristics such as low sexual health literacy that may be associated with both. To this extent, there remains an even more substantial gap in literature on PrEP-related adherence along the HIV prevention continuum, which involves pathways flowing from linkages to prevention services, retention in these services, and adherence support [13]. Our study seeks to address this gap in literature by understanding the multi-level factors influencing HIV risk behaviors and oral PrEP use among Black and Latino men who have heterosexual contact in NYC. The study elicits voices from both English speakers and LEP Spanish-speaking men. Given the U.S. States federal government's commitment to reducing new HIV infections by 90% by 2030, as well as the continued roll-out of newer long-acting injectable PrEP, it is imperative that HIV prevention efforts equitably understand factors impacting HIV prevention efforts among at-risk, and underserved populations [14,15].

## Methods

### Participants

This qualitative study was part of a larger clinical trial (n = 200) to evaluate the effectiveness of an intervention to increase oral PrEP initiation and retention among HIV negative adult patients accessing care at [blinded for review] Hospital. Participants provided written consent. Study participants were enrolled from June 19, 2019 through January 21, 2022. [Blinded for review] Hospital is part of a larger network of safety net hospitals providing care for underserved New Yorkers. The intervention component of the larger clinical trial involved recruiting individuals from various departments within the hospital (e.g., Obstetrics/Gynecology, Emergency Room) into the Sexual Health Clinic with the assistance of on-site navigators. On-site navigators reviewed the electronic health record of individuals admitted to the emergency room and screened for those who tested positive for a sexually transmitted infection. Additionally, participants were provided with accelerated follow-up with PrEP providers, point-of-care counseling, lab testing, and longitudinal care. Participants for the larger clinical trial were eligible if they were HIV negative and indicated for PrEP as assessed by the Sexual Health Clinic providers utilizing the Centers for Disease Control & Prevention guidelines [16]. For the qualitative study, a subset of this larger cohort was approached using a convenience sampling approach. Participants of the qualitative study were eligible if they identified as a cisgender man, as Hispanic/Latino or Black, engaged in heterosexual contact at the time of recruitment, and spoke English or Spanish. Participants were interviewed at any time point during the continuum of care: immediately after study enrollment and after having received point-of-care counseling and lab testing (i.e., some participants were prescribed PrEP on the same day of the interview) or any time point thereafter (e.g., on PrEP, prescribed PrEP but declined, discontinued PrEP).

### Procedures

In-depth interviews elicited information on HIV risk behaviors and barriers and facilitators to oral PrEP use. Semi-structured interview guides were developed based on a wide-ranging factors influencing HIV risk behaviors and PrEP use based on the literature. The interview guide included sections on social determinants of health (e.g., housing, employment), sexual partners and behaviors (e.g., intimate partner violence, concurrent partnerships, condom use), and PrEP-specific barriers and facilitators. All interviews were conducted in English or Spanish, audio-recorded, transcribed

verbatim, and translated if completed in Spanish. Interviews ranged from 30 minutes to an hour, and were conducted in-person at the hospital or by phone. The interviewing team was comprised of three trained qualitative interviewers: the first author (blinded for review) is an Asian, gender non-conforming individual with several years of experience conducting graduate-level qualitative research. The second interviewer (second author; blinded for review) is a Latina cisgender woman with several years of experience conducting qualitative research at the post-graduate level, and is fluent in Spanish. The third interviewer (last author; blinded for review) is an Asian cisgender woman with over 10 years of experience conducting and teaching qualitative research and is fluent in Spanish. The first and second interviewers were trained and supervised by the last author. Participants were provided with a $25 gift card upon completion. Interviews were conducted between October 2021 and December 2022. The study was approved by the Institutional Review Board of the [blinded for review].

## Data analysis

Data were iteratively analyzed using an inductive process integrating narrative and thematic approaches [17]. The transcripts were double coded and a codebook was iteratively developed by a team of two coders that included two of the interviewers (first author and second author; Blinded for review). Reliability of coding was ensured through frequent discussions and consensus between coders to achieve inter-coder reliability [18]. After reading transcripts multiple times, the study team compiled several themes and subthemes across participants, utilizing each participants' narratives to ensure that responses were grounded in their specific contexts. Our analytical approach aligns with Dr. Lisa Bowleg's (a Black feminist and intersectional scholar) approach to qualitative analysis [19], which involves overlaying structural and historical knowledge of the authors to interpret participant narratives and make connections between these structural factors and downstream health-related decision-making. The coding and analyses were conducted via Atlas.ti, a software package for qualitative data analysis [20].

## Results

### General characteristics of study participants

Sixteen individuals were interviewed. Participant characteristics are presented in Table 1. The median age of participants was 38.5 years old, with a range from 29 to 59 years of age. A majority of participants identified as Latino (n = 11, 68.75%). Five interviews with Latino participants were conducted in Spanish, while the rest were conducted in English. The results are organized into the following themes relating to multilevel factors on oral PrEP and HIV risk behaviors: 1) conceptualization of HIV risk behaviors; 2) attitudes towards PrEP among participants and their social network; 3) hegemonic masculinity and mixed, diverse sexual networks; and 4) social and contextual factors. Fig 1 depicts these main themes, which were not mutually exclusive and instead informed one another. This figure demonstrates how factors influencing both HIV risk behaviors and oral PrEP use share characteristics, such as a participants' beliefs around their own susceptibility to HIV.

**Table 1. Sociodemographic characteristics among Black and Latino Men with heterosexual contact in New York City, NY (n = 16).**

|  | Median (Range) or N (%) |
| --- | --- |
| Age | 38.5 (29–59) |
| Race<br>Black or African American<br>Latino | 5 (31.25%)<br>11 (68.75%) |
| Language of Interview<br>English<br>Spanish | 11 (68.75%)<br>5 (31.25%) |

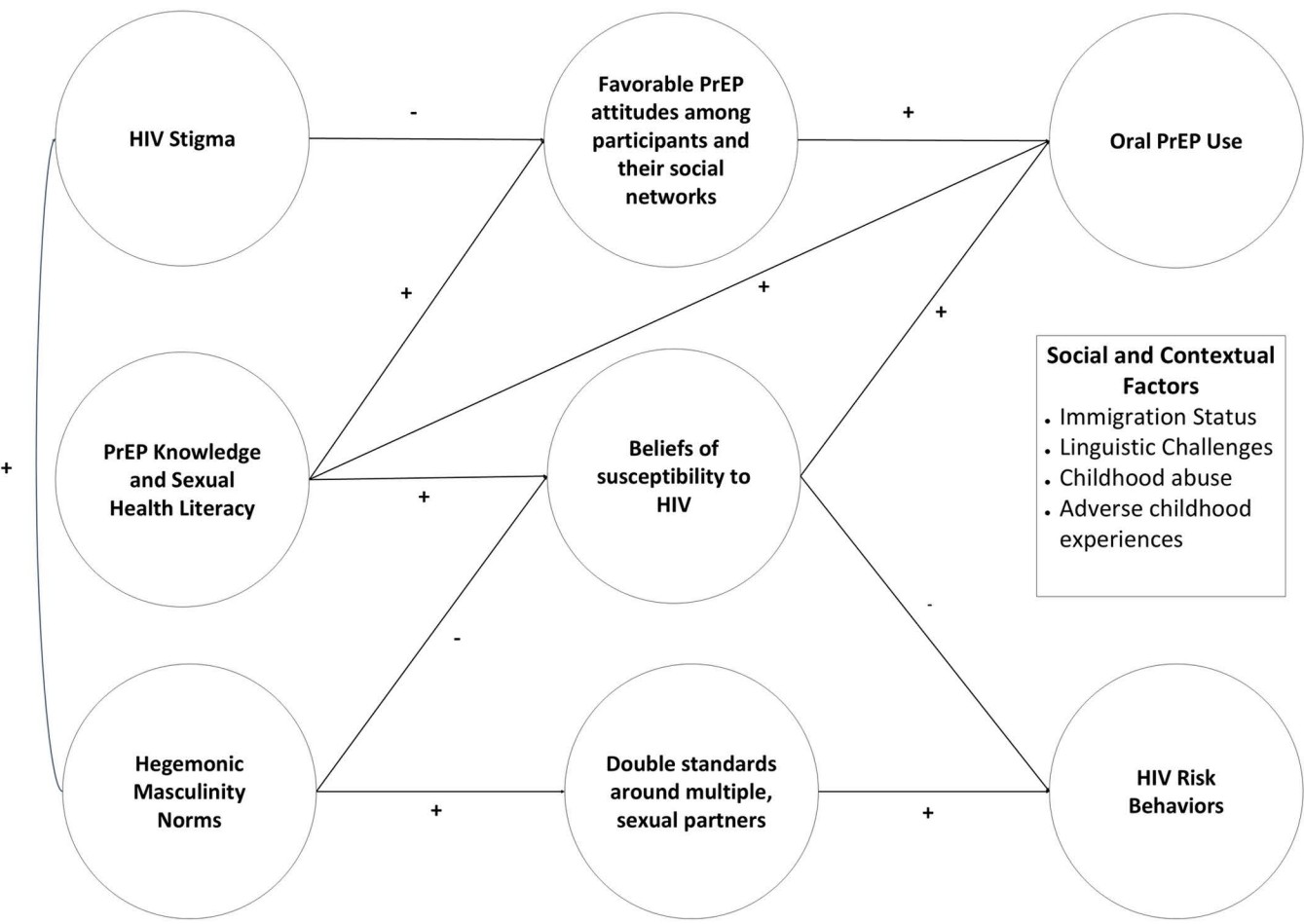

**Fig 1. Multilevel factors impacting HIV risk behaviors and oral PrEP use.**

## Theme 1: Conceptualization of HIV risk behaviors

Men's HIV risk behaviors were deeply informed by their own perception of sexual risk, stigma towards HIV, and understanding of PrEP.

**Self-perception of sexual risk.** Most men did not conceptualize their own sexual behaviors as risky. Rather, participants contextualized risk of infection as deeply rooted in trust, or lack thereof, towards sexual partners' disclosure of HIV risk behaviors. Infection with an STI or HIV was attributed to individuals making the wrong decisions about who to trust, as one participant noted when asked why they thought one of their former sexual partners had contracted HIV: "Made a mistake? That's probably it. Trusting people. Made a mistake." (English-speaking, Latino participant). In another instance, one participant noted the uncertainty he felt due to the lack of trust he had towards his perception of potential sex partners' HIV risk behaviors:

> So, when I'm speaking to them, I'm finding out…a woman could be thinking I'm talking to her to do other things. But I'm doing my homework. You know what I'm saying? From the way she talks to the way she answers my questions to everything. So, you know what I mean? From everything. I have to know. I have to have a system mind to be like, okay, she's been here before. She does this or she doesn't do this regular. Or – you know. It's just certain things that I go with and certain things that I don't… (English-speaking, Black participant).

Personal levels of trust towards intimate partners' HIV risk behaviors governed both an individual's understanding of their own risk of contracting HIV and their comfortability with engaging in condomless sexual intercourse. Combined with relatively low levels of sexual health literacy, a majority of participants had indicated that they did not feel at risk for contracting HIV. One participant, who traveled to the Dominican Republic and engaged with multiple sexual partners, mentioned that he just "felt comfortable, I was introduced through family. You know, her, she's clean. She's not like that… Meaning she's not like a, she's not promiscuous, she's not, you know? But everybody lies and I figure that out later on." (English-speaking, Latino participant 79). The participant seems to detach his own sexual behaviors from his HIV risk, emphasizing that the chance of contracting HIV and STIs is impacted not by his own behavior but rather a combination of trust towards partners and their partners' sexual histories. To this end, this participant did express some levels of perpetual distrust towards his intimate partners, suggesting that personal judgment alone is insufficient in fully trusting a sexual partner. Yet, these levels of distrust—particularly around whether partners accurately conveyed their own risk behaviors—did not change his sexual behavior.

One implication of men's utilization of trust as a measure of HIV and STI risk was that condom use was often left to snap judgements and personal intuition, instead of informed decision making through a reflection of one's own sexual histories and behaviors:

> I met her somewhere, like through a friend. So I have her like phone number still, like, you know, but we still like apart right now, but I like, I felt comfortable for some reason. I don't know why I just felt comfortable with her, but cause I knew her like already a day before I knew her. (English-speaking, Latino participant).

A few participants were able to identify that their behaviors were impactful in assessing their own risk level, as one participant frames their own understanding of risk: "First of all, monogamous, safe. Basically abstinence, we don't really engage. I don't go outside. I don't do anything really, as far as that's concerned." (English-speaking, Latino participant).

**HIV-related stigma.** Stigma around HIV was intertwined with participants' own perception of sexual risk and their HIV risk behaviors. Participants often othered individuals living with HIV or characterized certain subpopulations as inherently HIV-infected, rather than recognizing the different degrees of HIV risk behaviors associated with various forms of heterosexual contact. When asked about his own risk of getting HIV, one participant answered:

> Like, on my behalf, I'm not worrying about catching HIV because I'm not out there. You know what I'm saying? I'm not doing certain things. I'm not sharing needles. I'm not doing nothing that has to, that I have to be worried about that. (English-speaking, Latino participant)

On one hand, the participant understands the increased risk for HIV transmission among injection-drug users. On the other hand, the participant feels that he is at minimal risk for catching HIV because he does not traditionally fall into "high-risk" groups. However, the same participant was indicated for PrEP because he had a prior STI diagnosis but did not conceptualize himself as belonging to a high-risk group.

In another instance, one participant described individuals with HIV as unhealthy and unable to participate in certain types of physical activity:

> I, I figured something out. See where, what I was telling you about Marcus Garvey [Park] on the top of the mountain, where the fire tower is… If you were sick, you wouldn't be able to climb that mountain. A sick person wouldn't be able to reach up there. Hmm, you understand? So that's why the guys that trust each other, you understand there's that level of trust because a person in bad health would never be able to make it to the top of that mountain. You understand? So I thought about why [men] do it there, because then I see a woman that's sick with HIV. (English-speaking, Black participant)

This characterization of individuals with HIV as sickly perpetuates the erroneous narrative that individuals with HIV are unable to live long, healthy lives. Moreover, the participant's misperception of s the sexual risk level of their behaviors is misguided by the fact that individuals with HIV, even when untreated, may present as relatively healthy early on in their infection.

**Understanding of PrEP.** A prominent sub-theme was related to a lack of understanding of correct PrEP use, which may contribute to increased HIV risk behaviors. During sexual intercourse, participants often expressed utilizing PrEP as an alternative to condoms, with few participants mentioning the use of both PrEP and condoms during intercourse. Moreover, participants repeatedly noted confusion around why they were prescribed PrEP, or did not identify PrEP as an HIV and not an STI prevention drug. Study participants were all men who met CDC criteria for consideration of PrEP initiationin accordance with CDC guidelines, with most participants being referred to the sexual health clinic from the emergency room after receiving positive STI tests. Despite this, when asked why they believed they were on PrEP, participants rarely mentioned that their past histories with STI diagnoses were indications for being prescribed PrEP. When asked why he wanted to take PrEP, one participant noted that he was unsure of what PrEP was for: "What is that medication for? Is it to prevent AIDS?...Well, I took it for about 15 days" (Spanish-speaking, Latino participant).

However, some participants did demonstrate their ability to understand on-demand PrEP (2 pills 2–24 hours before sex, 1 pill 24 hours after the first dose, and 1 pill 24 hours after the second dose), rather than daily PrEP intake. One participant described that they did not take PrEP pills daily but instead learned about on-demand PrEP from the physicians at the sexual health clinic:

> I wouldn't like to take [PrEP] because I am not at it [sex] every day. So, he told me that he was going to send me some pills that I could take just two hours before intercourse. He said that those would be better for me. (Spanish-speaking, Latino participant)

On-demand PrEP was often conceptualized as an alternative to daily PrEP uptake, though clinical guidelines underscore the importance of daily oral PrEP ingestion if possible—especially among individuals, including the study participants, who were indicated for PrEP use by sexual health clinic providers.

### Theme 2: Attitudes towards PrEP among participants and their social networks

Despite low PrEP knowledge, men generally demonstrated positive attitudes towards PrEP, whereas men's social networks consisted of individuals who lacked PrEP awareness or had markedly negative attitudes towards PrEP. One participant, when asked about their personal feelings towards PrEP, felt empowered by their use of PrEP and ability to be self-sufficient in their sexual health decision making:

> How do I think other people feel about PrEP? That's a good question. I think they should feel good about it. Shit. I think they should feel good about it. I don't know how they feel about it. But you gotta be honest with it. You gotta be honest. You gotta be honest with yourself and know everybody's not out talking-wise. … But I think people are just scared. But I think if they knew what it was and the risk that it could help them, I think they would be more open to it. (English-speaking, Black participant)

When individuals had an accurate understanding of PrEP, positive attitudes towards PrEP were even more pronounced. One participant noted that they felt protected from HIV because of PrEP's effectiveness:

> From what I've heard –I don't wanna jinx anything–my chances are very, very low - very low. God forbid if that condom were to break. It's very low…I'm not saying I'm an immortal, I'm not saying I'm God. I'm just saying I'm glad I'm on it. I'm glad this drug does exist. I'm glad it offers protection. (English-speaking, Latino participant).

While positive attitudes towards PrEP were prominent among participants, outside perceptions towards PrEP were often grounded in stigmatizing beliefs around HIV, promiscuity, infidelity, and homosexuality. One participant felt uncomfortable disclosing that they were taking PrEP to their intimate partners to avoid misunderstandings of their HIV-negative status: "They look it up and they think it's HIV medication. So I just keep it classified." (English speaking participant). In another instance, a participant indicated that admission of PrEP usage was synonymous with promiscuity: "I think it's kinda weird because I know what they [sex workers] do, but I just feel that if I would tell them, they would probably be like, 'You're a man whore or something.'" (English speaking, Latino participant). At the same time, outsider perspectives towards PrEP were also informed by historical stereotypes around HIV: "They say the same thing like o yeah, that's just for homosexuals, or that's for gay people. I don't wanna say they say it's only for them, but they feel like that's who the commercials and stuff are talking to, homosexuals" (English-speaking, Black participant).

In one participant, attitudes of PrEP from his girlfriend dampened his enthusiasm to adhere to daily PrEP use, as his partner explicitly mentioned that PrEP use is an indication of infidelity:

> Because my girl found it one time and she was like, ooh, so, you were like – it's complicated, man. If your girl sees you having PrEP, then she knows you're outside having fucking sex. And unprotected sex. So, it's a little complicated. So I asked him [the doctor] one day, is there a way to take it? Because it's not–you know what I mean? So you gotta think about it. If I'm not taking it every day and I go get it refilled…I have bottles of the shit sitting around. (English-speaking, Black participant).

### Theme 3: Hegemonic masculinity and mixed, diverse sexual networks

Hegemonic masculinity was embedded in virtually all participants' beliefs around STIs, HIV, and sexual behaviors. Men conformed to social norms around hegemonic masculinity through their misogynistic characterizations of women and double standards around multiple sexual partnerships. However, men also implicitly resisted notions of hegemonic masculinity by engaging in mixed, diverse sexual networks consisting of non-heterosexual contact and sexual intercourse with non-cisgender individuals.

***Misogynistic characterizations of women.*** Several participants viewed women as implicitly "adulterous" and/or incapable of engaging in protective sexual health behaviors. One participant outlines their thought process in determining whether they would engage in sexual intercourse with a woman:

> No, like, but same thing, like what I tell you, you don't know if a woman tricking you, even one you trust. And she never going to tell you I do this and do that…because you got your wife or the wife got the husband, they show a lot of love, but you don't know…Cause you never expect like your wife to go play around. (English speaking, Latino participant)

At the same time, men exhibited double standards around multiple partnerships. While women were scrutinized for being "tricky," men felt comfortable discussing their own sexual engagement with multiple women while being in a monogamous relationship. One participated shared that they were in a monogamous relationship with their girlfriend, but had difficulty settling down and instead engaged with multiple women at once:

> "It's not like, hey, I wanna settle down with you, girl. Let me see what you're about. Oh. We had sex. Oh. That sex was great. Let's settle down and go to the movies and let's go eat. I don't be trying to do none of that shit. I mean, it sounds good. And if she's cool in a person, then when I have time then we can do that. But to be honest with you, I don't – and even if we are doing that, I'm still looking at other women and I'm still trying to have sex with other women." (English-speaking, Black participant).

Within men's sexual networks, gender norms often dictated who participants perceived as "high-risk" sexual encounters. One participant—a recent undocumented immigrant from England—frequented a New York City park to receive sexual services and described a gendered hierarchy of sex workers based on his perception of STI and HIV risk:

> Well, the, her, when I came here for herpes, it was from a woman, not from a man…It was from a woman, and that was in Central Park and she was a prostitute, but I caught the herpes from a woman, not from a man. I didn't catch no disease from a man having sex with all these man and transvestites, I never get no disease from. It's coming from women. (English-speaking, Black participant)

Women were positioned as subordinate to men who have sex with men (MSM) and transgender individuals. Although this participant identifies as an individual who engages in heterosexual contact, he portrayed a clear hierarchy of individuals who are less likely to have STIs and HIV in comparison to women. In doing so, he reinforced existing stereotypes of women as somehow weaker, a stereotype that was compounded with stigmatizing views of HIV as a crippling illness. Despite the expectations put forth by participants onto women partners, participants expressed having concurrent, multiple sexual relationships with others while being married: "I had the one I broke up with and my wife. And the other day I met someone but things happened quickly. That was three weeks ago." (Spanish-speaking, Latino participant).

Hegemonic masculinity was also embedded in participants' workplaces, with one participant explaining that their first experience interacting with sex workers was at their job. The participant described the normalization of hiring sex workers after his coworkers and him were "finishing their jobs", a stereotype perpetuated and reinforced by his boss:

> The first time I ever did it was when I was working in a mechanic store. I was working in the mechanic shop, fixing cars. And then my, my boss, he was, you know, he always hired sex workers after they finished jobs, he always invited me. And I was like, nah, that's not me, that's not me. Until one time, I decided like, you know what, let me just do it. That was in 2014. Yeah, I remember that day…And then he would buy beers and then later on he would just sit down and then at the porch, and then later on he would hire prostitutes. (English-speaking, Latino participant).

**Mixed and diverse sexual networks.** Participants expressed engaging in mixed and diverse sexual networks, which involved participants having sex with multiple cisgender men, cisgender women, and transgender women. While all participants included were individuals who engaged in heterosexual contact, many participants shared that they had non-heterosexual contact with others:

> "See, but to be honest with you, I was having sex with more than one person recently. I'd say a few of them…to be honest you, see, I met a Puerto Rican man. I saw him one day in the front. I saw, I met a Puerto Rican guy one day and then, um, like the performed oral sex on me and he did it like maybe three days and I was meeting him every Friday. You see? But then at the same time with a black transvestite…I just like, like I could leave here today right now and pass through the park or meet someone other, and then go have sex. That's how I am." (English-speaking, Black participant).

In particular, several participants expressed explicit interest in having sex with transgender women: "And during that season, I had sex a few times with a transexual [woman] without a condom – like two or three times." (Spanish-speaking, Latino participant).

Some participants also had multiple, concurrent sexual relationships. In one instance, a participant discussed their decision around unprotected condom use in one of their three romantic relationships, but not with the others:

> "I used to have two, three girlfriends before but not anymore…I used to use condoms with everybody. When I met this lady, [redacted]. It was the same thing. We used condoms all the time. After we check up, do blood test, that we was dating normal and she was my girlfriend to marry her, that's why this time I have sex without condom." (English-speaking, Black participant).

 

Regardless of their own participation in non-heterosexual or non-cisgender sexual contact, participants voiced views that were implicitly and explicitly stigmatizing against the LGBTQ+ community and in-line with hegemonic masculine norms. In describing their experiences with transgender women, participants utilized vocabulary that was either outdated, like "transexual," or diverged from the latest terminology: "I find transvestites, sex with the transvestite…they understand more, they got better sense than women, honestly" (Spanish-speaking, Latino participant). When one participant, who engaged in sexual intercourse with a man, was asked whether he identified as gay, he responded: No, gay means like…gay means you like men, you know, see, I like women. I like transvestites. I like, I like everybody. So I don't, I don't see myself as I, I like gay people." (English-speaking, Black participant).

**Theme 4: Social and contextual factors**

A prominent theme among all participants was the existence of social and contextual factors such as past and current substance use, child adversity and intimate partner violence, socioeconomic barriers, immigration, and language barriers that may have posed as additional barriers in practicing protective HIV behaviors and in consistently using PrEP.

A few participants experienced childhood adversity in the form of childhood labor and sexual abuse, which likely informed HIV risk behaviors later in life. One participant clearly mentions his experiences of physical and sexual abuse in his childhood:

> Yes, it's about physical abuse like bullying at school. And also, physical abuse like getting hit by my father as a result of his alcoholism. He would come home and be very aggressive with us and my mother. We saw how he beat her. And there was also sexual abuse, we were innocent children, and there were adults around us that would touch us inappropriately. So, we had to grow up with this trauma. It wasn't just me. It was also another brother of mine. (Spanish-speaking, Latino participant).

Experiences of early childhood violence were especially prominent among immigrants, who sometimes normalized violence at home. The same participant, describing their childhood, notes: "I had some trauma like any other child. There was some abuse and other things that helped me become who I am and shape my character" (Spanish-speaking, Latino participant 35). Participants also explicitly disclosed experiencing adverse childhood events (ACEs), including witnessing violence between their parents as a child. One participant noted the lifecourse impact that these traumatic experiences had on his approach to relationships:

> "Yes, so I am one of those people who experienced my father abusing my mother when I was growing up. So, I am not about that. So, when I see a woman that is not well, I feel bad. So when I met her, she would tell me, 'This is what I am going through. I am being treated this way.' And I said, 'Look. Don't put up with that. I can help you.'" (Spanish-speaking, Latino participant).

Participants also showed a sustained pattern of using substances, which at times impacted their ability to engage in protective sexual behaviors:

> "Yes, I wore a condom. I did wear them most of the time except for a couple of times when I was under the effects of alcohol, and I wasn't being coherent. I wasn't physically well. I am referring to the fact that I was drunk. So, I didn't wear a condom a couple of times" (Spanish-speaking, Latino participant).

At times, substance use behaviors were initiated when participants experienced drastic life changes, such as one participant who started utilizing substances heavily after their move to the U.S. from Mexico ten years ago:

"I don't remember very well because I started using alcohol. I still drink but not like before. I started using drugs, and I was on the street, yes." (Spanish-speaking, Latino participant).

Additionally, participants expressed challenges in regards to their immigration status. One participant, a Black, Native American man who had immigrated from England, expressed compounding difficulties originating from their documentation status:

I'm in a man's shelter. [It's] for men that have substance abuse, alcoholism. And I've been there since January, before that I was sleeping in Morningside Park…right now I am not working because of identification. New York City makes it so difficult to have an ID. But when the police handcuff you, they can ID you. The cops can ID you, but the department of motor vehicles can't? Come on now, but that's what's holding me back from employment is having identification. (English-speaking, Black participant).

Finally, linguistic challenges were especially prominent among Spanish-speakers, who noted that language barriers extended beyond the workplace and into their healthcare systems: "As I mentioned, sometimes they would send me the results [STI testing], but they were in English, and I don't understand much English. So, I really didn't understand." (Spanish-speaking, Latino participant).

## Discussion

This study sought to qualitatively explore multi-level factors influencing oral PrEP uptake, adherence, and HIV risk behaviors among cisgender Black and Latino men who engage in heterosexual contact and receive care at an urban safety net hospital in New York City. In addition to broadly exploring barriers and facilitators to oral PrEP uptake and adherence, interviews revealed men's interactions with their mixed and diverse sexual networks, which were informed by hegemonic masculinity [21–23]. HIV risk behaviors were also contextualized through participants' experiences with social and structural factors such as substance use, trauma, and language barriers across their lives, indicating a need to better frame HIV risk through a trauma-informed, life course lens [24,25].

Partner- and gender-related factors were among the most prominent themes and sub-themes and impacted participants' HIV risk behaviors and romantic and/or sexual networks. Women were often regarded as either inherently untrustworthy or riskier sexual partners, which thus influenced participants' willingness to utilize protection or engage in sexual intercourse with women. HIV risk behaviors were deeply embedded in hegemonic masculine norms, ones which position women as subordinate with little power in matters involving sexual intercourse and relational decision making [23]. Relational power is a significant predictor of condom usage among women, with women with high levels of relationship power reporting higher levels of consistent condom use than those with low levels of relationship power [26]. Interventions should consider incorporating discussions of sexual health and decision at the interpersonal level between men and their partner, which in turn may contribute to broader societal efforts to address gender equity [7]. As both gendered norms and men's perception of trustworthiness were critical factors for decisions around sexual intercourse and condom usage, providers may benefit from further educating men on their individual factors that increase HIV and STI transmission risk. For example, all men in the study were indicated for PrEP by their primary care providers, but most men did not explicitly make connections between their past STI histories and increased transmission risk.

Additionally, a closer inspection around the processes of hegemonic masculinity among US men may produce narratives that enhance conversations on how men resist and/or conform to gendered norms influencing HIV risk behaviors [27,28]. In particular, themes around hegemonic masculinity were juxtaposed with participants' willingness to engage in and disclose information about concurrent sexual and romantic relationships with gay men, bisexual men, transgender individuals, and sex workers. Participant's mixed sexual networks demonstrate that men with heterosexual contact may critically challenge societal assumptions that reinforce norms around sexual dominance over women. To this extent,

participants were comfortable disclosing the mixed nature of their sexual and romantic networks, a notion to the idea that men are open to discussing sexual health behaviors antithetical to the pressures put forth by hegemonic masculinity [29]. Despite engaging in these diverse and mixed sexual networks, participants perpetuated stigmatizing attitudes towards the LGBTQ+ community, utilizing inconsistent terminology when referring to transgender populations and exhibiting compensatory behaviors to resist notions that they are gay. The study calls for more research in regards to these mixed sexual networks, as a lack of understanding around HIV risk behaviors among men with heterosexual contact in the U.S. may grossly underestimate the true transmission risk of STIs and HIV for this population. At the patient-provider level, providers are encouraged to directly ask patients about the nature of patients' relationships, with an emphasis on avoiding gendered stereotypes of men who engage in heterosexual contact.

At the individual level, participants expressed limited health literacy around HIV. Men underscored their belief that they were not as at risk as sexual minority men and implicated HIV as a "gay disease." The association of HIV with certain subpopulations is a historically-consistent paradigm, one that continues to stigmatize LGBTQ+ (Lesbian, gay, bisexual, transgender, queer, and other) populations, immigrant communities, and communities of color [30,31]. This paradigm was further reinforced through targeted advertisements of PrEP on social media and television, which participants noted were only directed towards men who have sex with men. Such stigmatizing beliefs are not only damaging to individuals with sexual minority identities but also inaccurate: numerous participants reported both heterosexual and non-heterosexual contact embedded in mixed sexual networks, which likely increased their risk of contracting HIV [32,33]. The dichotomization of men with heterosexual contact vs. sexual minority men fails to reflect the realities of sexual behavior among individuals with mixed sexual networks, and future research may benefit from interrogating whether current characterizations of these subpopulations are accurate.

Participants also exhibited limited understanding of PrEP use, with few describing the importance of consistent ingestion of oral PrEP and dual usage of PrEP and condoms. The dual usage of PrEP and condoms serves to provide the most comprehensive protection against HIV and STIs together and should be repeatedly emphasized as best practice to prevent transmission [34,35]. Of note, several participants demonstrated an understanding of on-demand PrEP, where participants would take oral PrEP before, immediately prior, and after sexual intercourse to reduce HIV transmission [36]. On-demand PrEP, if taken correctly, is in line with harm reduction principles, which aim to reduce the negative consequences of sexual intercourse [37,38]. As such, providers should continue to educate their participants on the consistent ingestion of oral PrEP and usage of condoms, while providing information on the potential utility of on-demand PrEP if taken as directed. for situations when participants may be unable to take oral PrEP daily.

In comparison to their limited understanding of PrEP participants overwhelmingly expressed positive attitudes towards PrEP. This finding may be unusual, as these positive attitudes deviate from behavioral health theories and understandings of hegemonic masculinity, which would suggest high levels of stigma and negative perceptions of PrEP as a medication "traditionally" designed for gay men who have sex with men. In this regard, future research may explore how hegemonic masculinity impacts oral PrEP use and related HIV risk behaviors among heterosexual men in the US context, as much of the extant literature has focused on countries such as South Africa with historically high rates of heterosexual HIV transmission [27]. While the study did not explicitly explore patient-provider factors impacting PrEP attitudes, positive PrEP attitudes may be driven by a welcoming, non-stigmatizing environment bolstered by positive patient-provider interactions in the clinic where this study was situated. Regardless of PrEP adherence, participants continued to return to this clinic and noted the role of the primary care providers in delivering patient-affirming care.

Finally, while this study focused on factors influencing oral PrEP uptake, adherence, and HIV risk, men openly disclosed their experience with sexual trauma when probed on their upbringings. Participants directly linked their sexual trauma and attitudes towards relationships, with one participant noting that they found solidarity with romantic and sexual partners who shared similar pasts. Disclosure of sexual trauma among men is a narrative that contradicts current understandings of the impact of trauma on men, as most literature has focused on the cycle of abuse: men experience physical

abuse during their childhood and later perpetrate physical abuse themselves [39]. The ability to openly discuss sexual trauma provides opportunities to investigate the impact of sexual trauma throughout men's lives and reject gendered norms suggesting men's inability to talk about sexual victimization [40–42]. Additionally, this disclosure of men's trauma may inform HIV risk behaviors, as extant literature underscores the strong association between traumatic experiences and the use of substances as a coping mechanism [43]. In turn, substance use may increase the likelihood of engagement in HIV risk behaviors [44]. Care may be improved if providers considered the impact of sexual trauma on men such as by embracing the best practices of trauma-informed care [45], as the life course impacts of such violence may influence HIV risk behaviors and interpersonal relationships. In this process, providers may benefit from receiving guidance around the best practices of trauma-informed care focused on providing patient-centered care and minimizing the potential of harmful interactions in health-care settings in the process of trauma disclosure [45].

This study has several limitations. While authors were able to observe differences across English and Spanish-speaking participants, language of interview is a proxy for immigration status and acculturation. Rather, future studies may benefit from utilizing a socioecological framework to better capture the various dimensions of individuals (e.g., religious, linguistic, cultural, political) and environmental factors across subgroups [46,47]. Our study could have also benefited from the application of an explicit intersectional framework to inform the development of the interview guide, as the inclusion of Spanish-speaking participants lends itself to an analysis of how multiple marginalization produces compounding outcomes for individuals—especially for those who may be immigrants with limited English proficiency and low socioeconomic status. Further, our study population is one that is already engaged in care and may miss or underreport important vulnerabilities experienced by populations who have challenges in accessing and engaging in care.

In spite of these limitations, our study has several strengths. While previous studies on oral PrEP have primarily focused on men who have sex with men, transgender women of color, or individuals who inject substances, our study elicits the voices of cisgender Black and Latino men with heterosexual contact, which includes immigrants and limited English proficient Spanish speakers. Our analysis highlights how hegemonic masculinity and experiences across the life course shape the role of communal and societal norms in influencing men's HIV risk behaviors and oral PrEP usage. Our results may facilitate HIV prevention by providing insights that increase effective use of oral PrEP and that reinforce the importance of multilevel interventions at the patient-provider, interpersonal, and communal level towards e fairer gender norms, greater self-sufficiency for sexual health decisions, and improved health literacy. Overall, this study fills an important gap in literature on oral PrEP use among men with heterosexual contact in the U.S., and future PrEP research and interventions should continue to include this population so as to reach the U.S.'s collective goal of ending the HIV epidemic by 2030 [15].

## Supporting information

**S1 File. Participant In-depth Interview Guide.**
(DOCX)

## Author contributions

**Conceptualization:** Ronald S. Braithwaite, Robert A. Pitts, Sahnah Lim.

**Formal analysis:** Ohshue S. Gatanaga, Dalila Victoria Lanza, Sahnah Lim.

**Funding acquisition:** Robert A. Pitts.

**Project administration:** Robert A. Pitts.

**Supervision:** Ronald S. Braithwaite, Sahnah Lim.

**Writing – original draft:** Ohshue S. Gatanaga, Dalila Victoria Lanza.

**Writing – review & editing:** Ronald S. Braithwaite, Robert A. Pitts, Sahnah Lim.

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
