## [Decision Letter · Decision Letter 0]

PONE-D-24-18209

Multi-level factors influencing HIV risk behaviors and oral PrEP use among Black and Latino men with heterosexual contact in New York City

PLOS ONE

Dear Dr. Lim,

Thank you for submitting your manuscript to PLOS ONE. After careful consideration, we feel that it has merit but does not fully meet PLOS ONE’s publication criteria as it currently stands. Therefore, we invite you to submit a revised version of the manuscript that addresses the points raised during the review process.

Please attend to both reviewers comments in full in your resubmission. Please also note that all underlying data in the manuscript must be made available in a publicly accessible repository prior to publication. These data should be deidentified (pseudonymised) to comply with your IRB/ethical approvals.

Note from the Editorial Office: Please note reviewer 2 accidentally uploaded a draft version of their comments (file name: "Reviewers comments.docx"). We have attached the final version of their comments as the file "Reviewer 2 comments_PONE-D-24-18209.docx". 

We look forward to receiving your revised manuscript.

Kind regards,

Sebastian Suarez Fuller, PhD

Academic Editor

PLOS ONE

Journal Requirements:

[This work was supported by Gilead Sciences Inc under Grant IN-US-276-5456 and by the NIH – National Heart, Lung, and Blood Institute [grant number 1R01HL160324].].

Reviewers' comments:

Reviewer's Responses to Questions

**Comments to the Author**

1. Is the manuscript technically sound, and do the data support the conclusions?

Reviewer #1: Partly

Reviewer #2: Yes

2. Has the statistical analysis been performed appropriately and rigorously? 

Reviewer #1: N/A

Reviewer #2: Yes

3. Have the authors made all data underlying the findings in their manuscript fully available?

Reviewer #1: No

Reviewer #2: No

4. Is the manuscript presented in an intelligible fashion and written in standard English?

Reviewer #1: Yes

Reviewer #2: No

5. Review Comments to the Author

Reviewer #1: This qualitative study reports on findings based on n=16 interviews conducted with individuals participating in a PrEP trial. While this is an important topic, several concerns diminshed enthusiam for the paper in its current form:

1. Potential bias of the researchers leading to concerns about how data was collected and analyzed,

2. Unclear if the sample size was sufficient to reach saturation on these primary themes among a heterogenous sample of LEP and English speaking Black and Latino men,

3. Characterization of interviewee participants is lacking leading to low generalizability of results.

4. Missed opportunities here to extend the research especially regarding how either of these groups of heterosexual men (Latino, Black, LEP) view PrEP in their individual contexts.

Specific comments below:

Abstract

• Background: Make clear that this is a heterogenous sample of LEP men and English-speaking men. The way it is written currently led me to believe on first read that the sample was all LEP black and Latino men but after reading to the methods within the paper, I now realize it is a mix of English-speaking and LEP men. I think this should be clear as some of the barriers faced to PrEP care for LEP men likely differ from English-speaking men.

• Inclusive language, “substance use problems” in results section should just be substance use or negative consequences due to substance use depending on what we are trying to get at here.

• Conclusion should be more concrete towards the results and not a call or discussion of implications.

Introduction

• The introduction is beautifully written

• Page 3, lines 55-64, it might be helpful to include the clinical guidelines for PrEP for heterosexual men or people in general. US Public Health Service: PREEXPOSURE PROPHYLAXIS FOR THE PREVENTION OF HIV INFECTION IN THE UNITED STATES – 2021 UPDATE, A CLINICAL PRACTICE GUIDELINE (cdc.gov)

• Page 3 lines 65-68, please make clear here that a similar shared characteristic among the population you interviewed were LEP.

• Page 4 line 72, please make clear here if the entire interview sample are LEP or if it is a heterogenous group of LEP heterosexual men and English-speaking men.

• Page 4 72-73 “U.S. States federal government” is awkward phrasing. Are you saying the State government of NY or the federal government of the United States.

Method

• The inclusion/exclusion criteria for both the parent study and the qualitative study presented in this manuscript should be stated much more clearly. For example, Page 4, Line 87-88 “indicated for PrEP” does this mean they were approached about PrEP and seemed interested or was this that they seemed to meet the clinical guidelines to be prescribed PrEP?. Additionally Page 4, Line 90-92: “…, engaged in heterosexual contact at the time of recruitment,..” Within what time period?

• Some of the information about recruitment for the larger study and this study are unclear. For example, Page 4, lines 82-86, participants of the larger trial were recruited from various departments within the hospital. Then the next sentence says they were identified through the emergency room. Which was it? Clarify.

• As many of these participants may have been at different places within their larger trial participation and that this trial participation seems to have convinced at least some to try or get on PrEP, it seems important to detail where each participant was on the trial continuum. Should be added to a Table 1.

• Include the interview guide as an appendix

• Data analysis: Was the codebook closed or open? Could new codes be added during the coding process? If they were, how did you handle it?

• What did the study team do to address and mitigate their own biases within the data collection and analysis phases of this study?

• Is there any information available as to how study investigators decided thematic saturation had been reached?

Results

• It’d be nice to know where each participant was on the trial continuum if that information is available. I think it’s important contextual information to characterize participants, especially if any of them got on PrEP during the course of the trial, as stated in previous comment.

• Is there more demographic information available for these participants? For example, Page 8, Lines 193-195 references they aren’t sharing needles. Do you have any further information to better characterize this group of black/and Latino English speaking and LEP men like percentage that are PWID, HIV risk behaviors, others to better characterize participants?

• Further, is there any information as to how many potential participants were approached yet refused? This is important for helping the reader understand the generalizability of the sample.

• Page 7 Lines 166-172: Some of this feels like participant blaming. Is there a better framing maybe more towards low levels of health literacy and knowledge about how their own behaviors influence risk?

• Page 9 Line 212 to 215: This again reads like blaming the participant but there are many contextual reasons that this happens that is beyond the participants control and is rooted in the contextual experience within which this man lives. Calling this misperception “misguided” feels like it belittles the contextual factors within which Black and Latino men exist like masculinity which, despite its effect on women comes with its own negatives for men. This leads to questions about bias influencing the study.

• Page 9 Lines 216-227 illustrates why there needs to be further characterization of these men. Were the majority of participants on PrEP after the intervention? I think this holds implications for the understanding of this research.

• Page 12 Line 284 – 286: Please find an alternative language to “double standards”.

• Page 12 Line 294-297: This quote does not seem to go with the written passage above it. Seems to me this person is saying that you never know who you can trust, even your wife or husband can be in an adulterous relationship. I don’t read this as viewing women as “implicitly adulterous”.

• The authors should adjust their tone when discussing their participants who trusted them enough to participate in interviews and share their stories. The current tone feels laden with bias against participants who agreed to be interviewed.

Discussion

• The last paragraph of this discussion would be strengthened by providing key conclusions/summary of findings.

Reviewer #2: Like I indicated in my comments, the authors need to edit the language including the grammar, typo errors and punctuation as well as explaining some of the quotes, which are not clear. I suggested putting an explanation in brackets again the quotes that are not clear.

6. PLOS authors have the option to publish the peer review history of their article (what does this mean? ). If published, this will include your full peer review and any attached files.

**Do you want your identity to be public for this peer review?** For information about this choice, including consent withdrawal, please see our Privacy Policy .

Reviewer #1: No

Reviewer #2: No

---

## [Author Response · Author response to Decision Letter 1]

22 May 2025

REVIEWER #1

Overall Comments

1. Potential bias of the researchers leading to concerns about how data was collected and analyzed.

We agree that bias is always a concern in regards to qualitative data collection and analysis, and have attempted to minimize such bias through a rigorous, team-based approach and have added this in the Data Analysis section:

“To address potential researcher bias in data collection and analysis, we employed reflexive practices throughout the study. During data collection, the use of a semi-structured guide allowed for consistency between interviewers while allowing participants to expand upon their own perspectives. Throughout data collection and analysis, team members engaged in regular debriefs and peer reviews of coding decisions to reflect on how positionality may influence the interpretation of responses and ensure that interpretations remained grounded in the data.

We also explicitly looked for and reframed stigmatizing language and/or language that feels laden with bias against participants.

2. Unclear if the sample size was sufficient to reach saturation on these primary themes among a heterogenous sample of LEP and English-speaking Black and Latino men,

We agree that the sample consists of a heterogenous sample and attempted to only highlight themes that reached saturation. The following statement was added in our discussion:

“While the smaller sample size may not fully capture the full range of experiences given the heterogeneity of LEP and English-speaking Black and Latino men, the depth and consistency of interview responses suggest that thematic saturation was achieved.”

3. Characterization of interviewee participants is lacking leading to low generalizability of results.

We included additional information in our Table 1 to provide the audience with additional characterization of our sample as allowed by in the consent document. It should be noted that the goal of qualitative research is not necessarily to generalize to a larger sample, but rather to gain a nuanced understanding directly from participants, particularly from those who are under-researched.

4. Missed opportunities here to extend the research especially regarding how either of these groups of heterosexual men (Latino, Black, LEP) view PrEP in their individual contexts.

Given our smaller sample size, we decided to focus on extending our research findings to common patterns we saw across our sample of Black and Latino men with heterosexual contact. We recognize this as an important starting point for understanding how PrEP is perceived within these communities. We hope our multiple calls to action regarding future research with larger, more segmented samples will result in research that is more tailored to individual contexts.

Abstract

• Background: Make clear that this is a heterogenous sample of LEP men and English-speaking men. The way it is written currently led me to believe on first read that the sample was all LEP black and Latino men but after reading to the methods within the paper, I now realize it is a mix of English-speaking and LEP men. I think this should be clear as some of the barriers faced to PrEP care for LEP men likely differ from English-speaking men.

We made this clearer in the abstract:

“This study sought to understand factors influencing HIV risk behaviors and oral PrEP use among a heterogenous sample of low income, limited English proficient (LEP) or English-speaking, Black and Latino men with heterosexual contact in New York City (NYC).”

• Inclusive language, “substance use problems” in results section should just be substance use or negative consequences due to substance use depending on what we are trying to get at here.

Thank you for pointing out that this language can be stigmatizing. We changed this to “substance use.”

• Conclusion should be more concrete towards the results and not a call or discussion of implications.

We strongly believe that having an implication of discussion/call to action is important given our research findings and have thus recategorized the Conclusion section as a “Discussion” section to make this clearer.

Introduction

• Page 3, lines 55-64, it might be helpful to include the clinical guidelines for PrEP for heterosexual men or people in general. US Public Health Service: PREEXPOSURE PROPHYLAXIS FOR THE PREVENTION OF HIV INFECTION IN THE UNITED STATES – 2021 UPDATE, A CLINICAL PRACTICE GUIDELINE (cdc.gov)

We agree that it would be helpful to lay-out what the clinical guidelines for PreP for heterosexual men are and have added the following information:

“In the same year, the CDC recommended the administration of PrEP to 624,000 heterosexual individuals in the U.S. with indications for PrEP including having vaginal or anal sex in the past six months with an HIV positive partner, or with one or more sexual partners of unknown HIV status, or having been diagnosed with a sexually-transmitted illness in the past six months”

• Page 3 lines 65-68, please make clear here that a similar shared characteristic among the population you interviewed were LEP.

We hope the clarification in the abstract has made it clearer that this is a heterogeneous sample that includes both LEP and English-Speaking individuals.

• Page 4 line 72, please make clear here if the entire interview sample are LEP or if it is a heterogenous group of LEP heterosexual men and English-speaking men.

Likewise, we have added clarification in the introduction to make this clear:

“The study elicits voices from a heterogenous sample that includes both English speakers and LEP Spanish-speaking men, as these are the most commonly spoken languages among Black and Latino men in NYC.”

• Page 4 72-73 “U.S. States federal government” is awkward phrasing. Are you saying the State government of NY or the federal government of the United States.

We agree that this phrasing is awkward or may add to confusion, and have changed it to:

“Given the U.S. federal government’s commitment to reducing new HIV infections by 90% by 2030, as well as the continued roll-out of newer long-acting injectable PrEP, it is imperative that HIV prevention efforts equitably understand factors impacting HIV prevention efforts among at-risk, and underserved populations

Method

• The inclusion/exclusion criteria for both the parent study and the qualitative study presented in this manuscript should be stated much more clearly. For example, Page 4, Line 87-88 “indicated for PrEP” does this mean they were approached about PrEP and seemed interested or was this that they seemed to meet the clinical guidelines to be prescribed PrEP?. Additionally Page 4, Line 90-92: “…, engaged in heterosexual contact at the time of recruitment,..” Within what time period?

We clarified the language to include that they were approached about PrEP according to clinical guidelines:

“Participants for the larger clinical trial were eligible if they were HIV negative and met the clinical guidelines for PrEP prescription as assessed by the Sexual Health Clinic providers utilizing the Centers for Disease Control & Prevention guidelines”

Additionally, we clarified that individuals were eligible if they reported heterosexual contact at their last sexual encounter:

“Participants of the qualitative study were eligible if they identified as a cisgender man, as Hispanic/Latino or Black, engaged in heterosexual contact at their last sexual encounter, and spoke English or Spanish.”

• Some of the information about recruitment for the larger study and this study are unclear. For example, Page 4, lines 82-86, participants of the larger trial were recruited from various departments within the hospital. Then the next sentence says they were identified through the emergency room. Which was it? Clarify.

We apologize for this confusion and have clarified that individuals were recruited from emergency departments.

• As many of these participants may have been at different places within their larger trial participation and that this trial participation seems to have convinced at least some to try or get on PrEP, it seems important to detail where each participant was on the trial continuum. Should be added to a Table 1.

While we have not included specific information on where individuals were in the trial, we included information regarding whether individuals were on PrEP at the time of the interview, when they had been first prescribed PrEP, and the date since the last PrEP refill in Table 1. We were unable to include any other additional information about participants along the trial continuum as our consent forms do not specify being able to elicit information beyond PrEP information from the larger trial.

• Include the interview guide as an appendix

We have attached the English version of the interview guide as an appendix.

• Data analysis: Was the codebook closed or open? Could new codes be added during the coding process? If they were, how did you handle it?

We included more description of the data coding process:

“The codebook was treated as an open, living document, and codes were added and redefined as new themes emerging from the data. Decisions to add codes were discussed regularly among the coders, and earlier transcripts were revisited when new codes were added to ensure uniform application of codes across the transcripts”

• What did the study team do to address and mitigate their own biases within the data collection and analysis phases of this study?

The study team consistently engaged in reflexive practices, and we agree that this is especially important to explicitly mention due to the nature of our research. We added the following statement:

“To address potential researcher bias in data collection and analysis, we employed reflexive practices throughout the study. During data collection, the use of a semi-structured guide allowed for consistency between interviewers while allowing participants to expand upon their own perspectives. Throughout data collection and analysis, team members engaged in regular debriefs and peer reviews of coding decisions to reflect on how positionality may influence the interpretation of responses and ensure that interpretations remained grounded in the data.”

• Is there any information available as to how study investigators decided thematic saturation had been reached?

We added how we determined thematic saturation in the procedures section:

“Interviews were conducted until reaching thematic saturation, which involved team members engaging in an iterative review of the data during the coding process to ensure no new themes or concepts were emerging, and existing codes sufficiently captured the responses across the participants.”

Results

• It’d be nice to know where each participant was on the trial continuum if that information is available. I think it’s important contextual information to characterize participants, especially if any of them got on PrEP during the course of the trial, as stated in previous comment.

We included information regarding PrEP prescription, and we were unable to include any other additional information about participants along the trial continuum as our consent forms do not specify being able to elicit information beyond PrEP information from the larger trial.

• Is there more demographic information available for these participants? For example, Page 8, Lines 193-195 references they aren’t sharing needles. Do you have any further information to better characterize this group of black/and Latino English speaking and LEP men like percentage that are PWID, HIV risk behaviors, others to better characterize participants?

Likewise, we have included as much information as we can as specified in our consent and IRB application process. We hope the addition of PrEP prescription information adds a little bit more context.

• Further, is there any information as to how many potential participants were approached yet refused? This is important for helping the reader understand the generalizability of the sample.

We used a convenience sampling approach, primarily driven by who the research team thought would be willing to participate and do not have any information regarding how many participants were approached yet refused. We recognize that this may add to some issues around generalizability, but we again believe that the purpose of qualitative research is not generalizability but to gain a nuanced understanding directly from participants, particularly from those who are under-researched.

• Page 7 Lines 166-172: Some of this feels like participant blaming. Is there a better framing maybe more towards low levels of health literacy and knowledge about how their own behaviors influence risk?

We reframed these findings to provide broader social context to avoid participant blaming:

“The participant appears to externalize his perception of HIV risk by focusing more on trust in his partners and uncertainty about their sexual histories, reflecting a broader dynamic in which risk assessment is shaped by limited access to accurate sexual health information and low levels of health literacy around his own sexual behaviors. To this end, this participant did express some levels of perpetual distrust towards his intimate partners outside of his personal assessment of sexual transmission risk with a given sexual partner—suggesting that personal judgment alone is insufficient in fully trusting a sexual partner.”

• Page 9 Line 212 to 215: This again reads like blaming the participant but there are many contextual reasons that this happens that is beyond the participants control and is rooted in the contextual experience within which this man lives. Calling this misperception “misguided” feels like it belittles the contextual factors within which Black and Latino men exist like masculinity which, despite its effect on women comes with its own negatives for men. This leads to questions about bias influencing the study.

We agree again that we always need to be contextualizing findings in terms of the societal forces that shape experiences for individuals. We have modified the language in accordance to this:

“This characterization of individuals with HIV as visibly “sick” perpetuates the erroneous narrative that individuals with HIV are unable to live long, healthy lives. This perception is deeply embedded in broader sociocultural narratives that influence how risk is understood and may lead to a misperception of an individual’s own sexual risk level, when in reality many individuals with HIV, even when untreated, may present as relatively healthy early on in their infection.”

• Page 9 Lines 216-227 illustrates why there needs to be further characterization of these men. Were the majority of participants on PrEP after the intervention? I think this holds implications for the understanding of this research.

Since the intervention was ongoing, we do not have information regarding whether the majority of participants were on PrEP after the intervention.

• Page 12 Line 284 – 286: Please find an alternative language to “double standards”.

We have described double standards in a clearer way to avoid confusion and misrepresenting what we mean:

“Men conformed to social norms around hegemonic masculinity through misogynistic characterizations of women and inconsistent expectations placed on women versus men regarding multiple sexual partnerships.”

• Page 12 Line 294-297: This quote does not seem to go with the written passage above it. Seems to me this person is saying that you never know who you can trust, even your wife or husband can be in an adulterous relationship. I don’t read this as viewing women as “implicitly adulterous”.

We agree that the quote does not align well with the theme and may be extrapolating inferences outside of what is written. We have decided to remove this quote.

• The authors should adjust their tone when discussing their participants who trusted them enough to participate in interviews and share their stories. The current tone feels laden with bias against participants who agreed to be interviewed.

We strongly agree with you that language should be used

---

## [Editor Report · Decision Letter 1]

Multi-level factors influencing HIV risk behaviors and oral PrEP use among Black and Latino men with heterosexual contact in New York City

PONE-D-24-18209R1

Dear Dr. Lim,

We’re pleased to inform you that your manuscript has been judged scientifically suitable for publication and will be formally accepted for publication once it meets all outstanding technical requirements.

Kind regards,

Sebastian Suarez Fuller, PhD

Academic Editor

PLOS ONE
---

## [Editor Report · Acceptance letter]

PONE-D-24-18209R1

PLOS ONE

Dear Dr. Lim,

I'm pleased to inform you that your manuscript has been deemed suitable for publication in PLOS ONE. Congratulations! Your manuscript is now being handed over to our production team.

Kind regards,

on behalf of

Dr. Sebastian Suarez Fuller

Academic Editor

PLOS ONE